# RETHINKING MIXTURE-OF-AGENTS: IS MIXING DIFFERENT LARGE LANGUAGE MODELS BENEFICIAL?

## ABSTRACT

Ensembling outputs from diverse sources is a straightforward yet effective approach to boost performance. Mixture-of-Agents (MoA) is one such popular ensemble method that aggregates outputs from multiple *different* Large Language Models (LLMs). This paper raises the question in the context of language models: is mixing different LLMs truly beneficial? We propose Self-MoA — an ensemble method that aggregates outputs from only the *single* top-performing LLM. Our extensive experiments reveal that, surprisingly, Self-MoA outperforms standard MoA that mixes different LLMs in a large number of scenarios: Self-MoA achieves 6.6% improvement over MoA on the AlpacaEval 2.0 benchmark, and an average of 3.8% improvement across various benchmarks, including MMLU, CRUX, and MATH. Applying Self-MoA to one of the top-ranking models in AlpacaEval 2.0 directly achieves the new state-of-the-art performance ranking $1^{st}$ on the leaderboard. To understand the effectiveness of Self-MoA, we systematically investigate the trade-off between diversity and quality of outputs under various MoA settings. We confirm that the MoA performance is rather sensitive to the quality, and mixing different LLMs often lowers the average quality of the models. To complement the study, we identify the scenarios where mixing different LLMs could be helpful. This paper further introduces a sequential version of self-MoA, that is capable of aggregating a large number of LLM outputs on-the-fly over multiple rounds, and is as effective as aggregating all outputs at once.

## 1 INTRODUCTION

Large language models have made remarkable strides in improving performance across different domains, with notable examples such as GPT (Achiam et al., 2023), Gemini (Team et al., 2023), and Claude (Anthropic, 2023). Significant efforts have been directed toward increasing model size and training data to boost capabilities. However, scaling at training time comes with steep costs, while scaling computation during inference remains largely underexplored.

A straightforward way to utilize test-time compute is ensembling, which aims to combine outputs of multiple LLMs (Wang et al., 2024a; Lin et al., 2024; Jiang et al., 2023a; Wang et al., 2024a). Among various ensembling approaches, Mixture-of-Agents (MoA) (Wang et al., 2024a) has garnered significant interest, achieving superior performance in challenging tasks such as instruction following (Wang et al., 2024a), summarization, data extraction (OpenPipe, 2024), and real-world code issue resolution (Zhang et al., 2024b). Specifically, MoA first queries multiple LLMs (proposers) to generate responses, and then uses an LLM (aggregator) to synthesize and summarize these responses into a high-quality response.

Previous research highlights the significance of model diversity within the proposers for optimizing the performance of MoA, primarily focusing on strategies for ensembling a diverse set of individual models. We consider **cross-model diversity** as the variation among different models. However, pursuing cross-model diversity may inadvertently include low-quality models, resulting in a quality-diversity trade-off. While previous studies mainly concentrate on achieving a high cross-model diversity (Wang et al., 2024a; Zhang et al., 2024b), we adopt a holistic perspective on model diversity by considering **in-model diversity**, which arises from the variability of multiple responses generated by the same model. In-model diversity enables us to aggregate multiple outputs from an individual model. Intuitively, leveraging outputs from the best-performing individual model can more effec-

tively navigate the quality-diversity trade-off by creating a higher-quality proposer mixture. Thus, we propose Self-MoA as depicted in Figure 1b, which utilizes the same prompting template as MoA but aggregates outputs that are repeatedly sampled from the same model, rather than from a set of different models. To distinguish, we use Mixed-MoA to refer to MoA configurations that combine different individual models when necessary.

Surprisingly, we find that Mixed-MoA is usually sub-optimal compared with Self-MoA, especially when there exist significant quality differences among the proposers. Specifically, we revisit the same experiment setting of MoA with six open-source instruction fine-tuned models as Wang et al. (2024a). Compared with Mixed-MoA which aggregates all six models, Self-MoA on the strongest model surpasses its mixed counterpart with merely half of the forward passes on the AlpacaEval 2.0 benchmark, showing a case of when intra-model diversity is more effective. Moreover, Self-MoA combined with two best-performed models on AlpacaEval 2.0 consistently achieves a 2-3 point gain and secures the top position on the leaderboard, which further confirms the effectiveness of Self-MoA in this evaluation task.

To explore the limits of model diversity for MoA, we extend our experiments to a setting with three specialized models, each excelling in a specific task. Specifically, we utilize Qwen2-7B-Instruct (Bai et al., 2023) for common sense QA (MMLU-redux (Gema et al., 2024)), Qwen2-Math-7B-Instruct (Bai et al., 2023) for mathematics (MATH (Hendrycks et al., 2020)), and DeepSeek-Coder-V2-Lite-Instruct (Zhu et al., 2024) for coding (CRUX (Gu et al., 2024)). We compare Self-MoA against a range of Mixed-MoA strategies, evaluating 13 combinations of individual models based on their average performance across the three tasks. Our findings indicate that, even in this promising scenario for Mixed-MoA where each individual model excels in a specific subtask, only two Mixed-MoA strategies slightly outperform Self-MoA by 0.17% and 0.35%. Furthermore, if we have prior knowledge of the tasks and employ task-specific models as proposers for Self-MoA such as DeepSeek-Coder-V2-Lite-Instruct on CRUX or Qwen2-Math-7B-Instruct on MATH, Self-MoA can significantly outperform the best Mixed-MoA.

To better understand the effectiveness of Self-MoA, we conduct a comprehensive investigation of the trade-off between quality and diversity in MoA, involving over 200 experiments. We use the Vendi Score (Dan Friedman & Dieng, 2023) to evaluate the diversity among the outputs of the proposers, while the average performance of the proposers serves as the measure of quality. In Section 3, we confirm that MoA performance has a positive correlation with both quality and diversity. Moreover, we clearly show a trade-off along the achievable Pareto front of quality and diversity. Interestingly, we find that MoA is quite sensitive to variations in quality, with optimal performance typically occurring in regions characterized by high quality and relatively low diversity. This finding naturally explains the effectiveness of Self-MoA, as it utilizes the strongest model as the proposer, ensuring high quality in its outputs.

Finally, we evaluate the performance of Self-MoA under increasing computational budgets. As the number of outputs grows, the scalability of Self-MoA becomes constrained by the context length of the aggregator. To address this issue, we introduce Self-MoA-Seq (Figure 1c), a sequential version that processes samples using a sliding window, allowing it to handle an arbitrary number of model outputs. Our findings show that Self-MoA-Seq performs at least as effectively as Self-MoA, enabling scalable ensembling for LLMs with shorter context lengths without compromising final performance.

Overall, our contributions are three-fold:

- We introduce Self-MoA, which leverages in-model diversity by synthesizing multiple outputs from the same model. Surprisingly, it demonstrates superior performance compared to existing Mixed-MoA approaches, which emphasize cross-model diversity, across a wide range of benchmarks.
- Through systematic experiments and statistical analysis, we uncover a core trade-off between diversity and quality among the proposers, emphasizing that MoA is highly sensitive to proposer quality. This finding also explains the success of Self-MoA, which leverages outputs from the highest-performing model, ensuring superior overall quality.
- We extend Self-MoA to its sequential version Self-MoA-Seq, which iteratively aggregates a small amount of outputs step by step. Self-MoA-Seq unlocks LLMs that are constrained by the context length and enables computation scaling during inference.

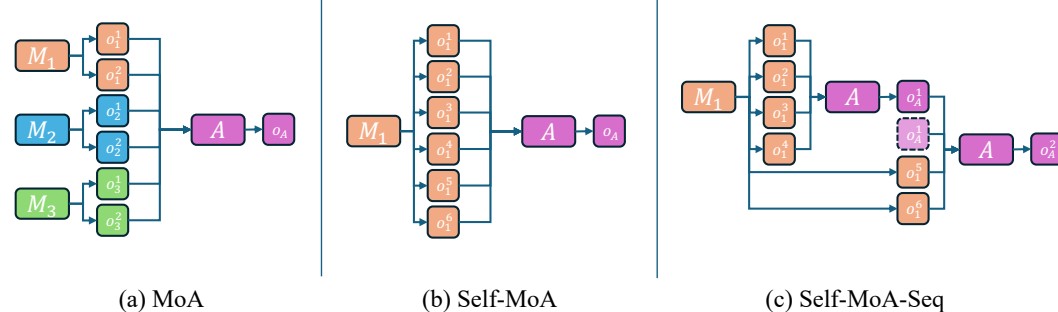

|  |  |  |
|:-:|:-:|:-:|
| (a) MoA | (b) Self-MoA | (c) Self-MoA-Seq |

Figure 1: Comparison of MoA, Self-MoA, and Self-MoA-Seq. (a) In MoA, multiple models respond to a query, followed by an aggregator synthesizing their outputs. (b) Self-MoA simplifies this by repeatedly sampling from a single model. (c) Self-MoA-Seq extends Self-MoA by applying a sliding window to combine the best output so far with candidate outputs. At each timestep, the synthesized output is repeated to bias the aggregator towards it, reducing the context length requirements and expanding the method's applicability. Note that MoA can extend to multiple rounds of aggregation (Appendix B.1), while Self-MoA and Self-MoA-Seq can extend to more outputs, but we omit them here for clarity.

## 2   IS ENSEMBLING DIFFERENT LLMs BENEFICIAL?

As introduced in Section 1, previous research primarily emphasizes **cross-model diversity**, which can inadvertently include low-quality proposers. In this work, we introduce Self-MoA (Figure 1), which uses a single top-performing model to generate multiple outputs and aggregate them to produce the final result. Self-MoA leverages **in-model diversity** as repeated sampling often produces varied outputs. We propose our research question as follows:

> *Does the benefit of MoA stem from cross-model diversity?*
> *Can we build a stronger MoA by utilizing in-model diversity?*

### 2.1   EXPERIMENTS ON ALPACAEVAL 2.0 WITH GENERAL PURPOSE MODELS

**Evaluation benchmarks.**   We adopt the same experiment setting as Wang et al. (2024a) in AlpacaEval 2.0 benchmark (Dubois et al., 2024) and compare the performance of MoA and Self-MoA[1]. AlpacaEval 2.0 is a widely used benchmark for assessing the instruction-following abilities of LLMs. It offers a set of real-world instructions and employs a GPT-4-based annotator to compare the model's responses against reference answers generated by GPT-4. To address length bias inherent in model-based evaluation, Dubois et al. (2024) introduced the length-controlled (LC) win rate as a more robust evaluation metric.

**Models.**   Following Wang et al. (2024a), we construct MoA based on six individual models: Qwen1.5-110B-Chat (Bai et al., 2023), Qwen1.5-72B-Chat (Bai et al., 2023), WizardLM-8x22B (Xu et al., 2023), LLaMA-3-70B-Instruct (Touvron et al., 2023), Mixtral-8x22B-Instruct-v0.1 (Jiang et al., 2024a), and dbrx-instruct (Team et al., 2024b). Each model is sampled with a temperature of 0.7, following the default in (Wang et al., 2024a). For Self-MoA, we aggregate six outputs sampled from WizardLM-2-8x22B, as it consistently outperforms the other models. In line with Wang et al. (2024a), we use Qwen1.5-110B-Chat as the aggregator for both MoA and Self-MoA.

**Results.**   We present the LC win rate for each model configuration in Table 1. For individual models, we report the higher value between the leaderboard results and our reproduction. Additionally, we include the total number of forward passes, where one forward pass is counted each

---

[1]We note that this experiment is similar to the "single-proposer" setting in Wang et al. (2024a), however our reproduced result is different. We conjecture that such a major difference is due to different choices of the proposer model, which is not mentioned in Wang et al. (2024a). As we shall see later in Section 3, ensembling performance is more sensitive to quality rather than diversity. Therefore, a worse proposer model will lead to suboptimal performance of Self-MoA.

Table 1: Comparison of Self-MoA and Mixed-MoA on AlpacaEval 2.0 leaderboard. We use Qwen1.5-110B-Chat as the aggregator.

|  | Model Configuration | LC Win Rate | # Forward Passes |
|---|---|---|---|
| Individual | WizardLM-2-8x22B | 53.1 | 1 |
|  | Qwen1.5-110B-Chat | 43.9 | 1 |
|  | LLaMA-3-70B-Instruct | 34.4 | 1 |
|  | Qwen1.5-72B-Chat | 36.6 | 1 |
|  | Mixtral-8x22B-Instruct-v0.1 | 30.2 | 1 |
|  | dbrx-instruct | 25.4 | 1 |
| Mixed-MoA | 2-Layer MoA (Wang et al., 2024a) | 59.1 | 7 |
|  | 3-Layer MoA (Wang et al., 2024a) | 65.4 | 13 |
| Self-MoA | 2-Layer Self-MoA + WizardLM-2-8x22B | **65.7** | 7 |

Table 2: Self-MoA achieves state-of-the-art performance on the AlpacaEval 2.0 leaderboard when using top-performing models as both proposers and aggregators. We only ensemble 4 outputs due to context window constraints.

|  | Model Configuration | LC Win Rate |
|---|---|---|
| Individual | gemma-2-9b-it-WPO-HB | 76.7 |
|  | gemma-2-9b-it-SimPO | 72.4 |
| Self-MoA | Self-MoA + gemma-2-9b-it-WPO-HB | **78.5 (rank #1)** |
|  | Self-MoA + gemma-2-9b-it-SimPO | 75.0 |

time a proposer model generates an output or an aggregator synthesizes a result. Notably, Self-MoA demonstrates remarkable effectiveness in this task, outperforming the strongest MoA baseline with only half the forward passes. This suggests that, while using multiple models intuitively offers greater diversity, ensembling multiple outputs from a single model is more effective.

**Applying Self-MoA on top performing models.** To further validate the effectiveness of Self-MoA, we apply it to the two top-performing models on AlpacaEval 2.0: gemma-2-9b-it-WPO-HB (Zhou et al., 2024) and gemma-2-9b-it-SimPO (Meng et al., 2024). We use each model as both the proposer and the aggregator[2], with a temperature of 0.7 for all the generations. Due to the context length constraint of Gemma 2 (Team et al., 2024a), the aggregator can only take four samples as the input. As shown in Table 2, Self-MoA consistently achieves a 2-3 point gain and secures the top position on the leaderboard during submission.

## 2.2 EXPERIMENTS ON MULTIPLE DATASETS WITH SPECIALIZED MODELS

In this section, we explore different ensembling methods on a diverse set of benchmarks using specialized models.

**Evaluation datasets.** We conduct evaluations across a diverse set of benchmarks:

- MMLU (Hendrycks et al., 2020) is a multiple-choice dataset designed to assess a model's multitask accuracy. MMLU is widely used to evaluate both the breadth and depth of language understanding capabilities of current LLMs across a diverse array of subjects, including mathematics, history, computer science, logic, and law. We adopt MMLU-redux (Gema et al., 2024) for evaluation, which is a subset of MMLU with 3,000 samples fixing the errors in the dataset through human re-annotating.

- CRUX (Gu et al., 2024) consists of 800 Python code functions, each containing 3 to 13 lines along with an input-output pair. Based on this dataset, Gu et al. (2024) constructs two tasks: input prediction and output prediction. To successfully complete these tasks, the LLM must demonstrate code reasoning abilities.

---

[2]Qwen1.5-110B-Chat is not used as the aggregator since the two top models significantly outperform it.

Table 3: Comparison of Self-MoA and Mixed-MoA in MMLU, CRUX, and MATH. Mixed-MoA models with top two average performances are highlighted by underline. The labels `i`, `m`, and `d` refer to Qwen2-7B-Instruct, DeepSeek-Coder-V2-Lite-Instruct, and Qwen2-Math-7B-Instruct, respectively. The average performance represents the mean accuracy across MMLU, CRUX, and MATH. `TaskBest` indicates that we use the strongest model for each task as both proposer and aggregator. For instance, in the case of CRUX, `TaskBest` refers to DeepSeek-Coder-V2-Lite-Instruct (`d`).

|  | Aggregator | Proposer | MMLU | CRUX | MATH | Average |
|---|---|---|---|---|---|---|
| Individual | - | i | 66.16 | 36.25 | 53.81 | 52.07 |
|  | - | d | 60.91 | 49.51 | 53.82 | 54.74 |
|  | - | m | 54.36 | 27.88 | 69.57[6] | 50.60 |
| Mixed-MoA | i | iimmdd | 67.89 | 42.88 | 64.38 | 58.38 |
|  |  | imdddd | 67.42 | 44.50 | 63.90 | 58.61 |
|  |  | iiiimd | 68.90 | 41.25 | 63.00 | 57.72 |
|  |  | immmmd | 66.63 | 42.75 | 66.02 | 58.47 |
|  |  | iimmmm | 66.23 | 39.25 | 66.10 | 57.19 |
|  |  | iiiimm | 67.49 | 38.25 | 64.16 | 56.63 |
|  |  | iiiimm | 68.00 | 37.00 | 62.92 | 55.97 |
|  |  | iidddd | 68.21 | 45.50 | 62.56 | 58.76 |
|  |  | iiiddd | 68.21 | 42.88 | 62.38 | 57.82 |
|  |  | iiiidd | 68.47 | 40.75 | 61.24 | 56.82 |
|  |  | mmdddd | 66.34 | 46.75 | 66.48 | 59.86 |
|  |  | mmmddd | 65.80 | 47.00 | 67.32 | 60.04 |
|  |  | mmmmdd | 65.44 | 42.50 | 67.62 | 58.52 |
| Self-MoA | i | dddddd | 65.23 | 50.75 | 63.08 | 59.69 |
|  | i | 6×TaskBest | **69.01** | 50.75 | 68.42 | 62.73 |
|  | TaskBest | 6×TaskBest | **69.01** | **52.62** | **69.80**[6] | **63.81** |

- MATH (Hendrycks et al., 2021) comprises 12,500 challenging competition-level mathematics problems. For our analysis, we utilize the testing subset of MATH, which consists of 5,000 samples.

**Models.** To ensure sufficient diversity, we select three LLMs with specialized strengths: Qwen2-7B-Instruct[3], DeepSeek-Coder-V2-Lite-Instruct[4], and Qwen2-Math-7B-Instruct[5]. We fix the number of proposers to six and sweep various combinations of these three models. For convenience, we denote Qwen2-7B-Instruct as `i`, DeepSeek-Coder-V2-Lite-Instruct as `d`, and Qwen2-Math-7B-Instruct as `m`. The evaluation results in Table 3 show that Qwen2-7B-Instruct, DeepSeek-Coder-V2-Lite-Instruct, and Qwen2-Math-7B-Instruct excel on MMLU, CRUX, and MATH, respectively. We use the short name for the mixture of proposers. For example, `iiddmm` indicates the inclusion of two samples from each model respectively. When a model is represented multiple times in the proposer mixture, we ensure that two samples are generated with different random seeds. We set the temperature of each model to be 0.7 for the individual model, and use temperature 0 for the aggregator. We mainly use Qwen2-7B-Instruct as the aggregator but also try different models as the aggregator. We explore various MoA configurations, including individual models, combinations of two or three models as proposers, and using a single model as the proposer (Self-MoA).

**Results.** The results are shown in Table 3. When considering `i` as the aggregator, among 11 tested combinations of proposers for MoA, only two combinations slightly outperformed Self-MoA with `dddddd`. Specifically, the combinations `mmdddd` and `mmmddd` outperformed `dddddd` by 0.17% and 0.35%, respectively. The performance of the remaining MoA models was inferior to that of `dddddd`.

---

[3] https://huggingface.co/Qwen/Qwen2-7B-Instruct
[4] https://huggingface.co/deepseek-ai/DeepSeek-Coder-V2-Lite-Instruct
[5] https://huggingface.co/Qwen/Qwen2-Math-7B-Instruct
[6] As Qwen2-Math-7B-Instruct only supports context length of 4096, for these two data points, we sample the proposer with a reduced token length of 1024, and only aggregates three outputs from the proposer.

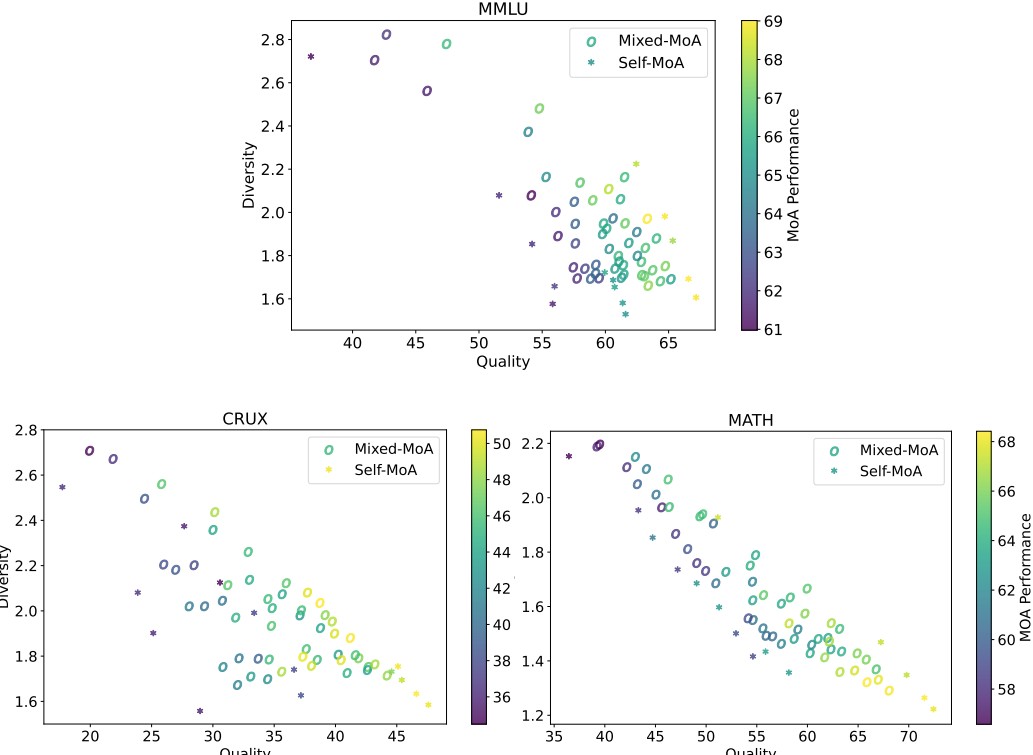

Figure 2: The diversity-quality trade-off: Mixed-MoA incorporates different individual models as proposers, while Self-MoA uses the same individual model for this role. Quality is assessed based on the average performance of each proposer, and diversity is computed with the Vendi Score (Dan Friedman & Dieng, 2023) of outputs generated by proposers on the same prompts.

Adding model diversity does not necessarily enhance performance. For instance, MoA with `iimmdd` performs worse than `mmmddd` in terms of average accuracy. Although model `i` is the strongest on MMLU among individual models, its inclusion in the proposers does not improve overall performance on the mixed datasets, i.e., `mmmddd` has 60.04% overall performance while `iimmdd` only has 58.38%.

The performance of Self-MoA can be significantly improved when we are allowed to select the strongest model for each task. This is particularly beneficial when we have prior knowledge of the task we wish to address. As shown in Table 3, when we use Qwen2-7B-Instruct as the aggregator, Self-MoA achieves a performance of 62.73% by selecting the appropriate proposer for each task. Additionally, employing a task-specific aggregator further boosts overall performance to 63.81%. We postpone more discussion to Section 3.2.

## 3 THE QUALITY-DIVERSITY TRADE-OFF

We investigate factors that contribute to the strong performance of Self-MoA through careful experiments. Previous studies have mainly focused on increasing model diversity within the group (Wang et al., 2024a; Jiang et al., 2023a; Zhang et al., 2024b). However, searching for diverse models can sometimes lead to including poorly performed models, resulting in a trade-off between diversity and quality, where quality refers to how well each individual model performs in the group.

Therefore, we aim to identify the existence of a general relationship between MoA's performance and quality as well as diversity. Following Section 2, we evaluate MoA's performance on MMLU, CRUX, and MATH, which cover tasks requiring a wide range of capabilities. We vary the quality and diversity with two orders of freedom: 1) combinations of individual models in proposers from Section 2.2; and 2) sampling temperature. i.e., 0.5, 0.7, 1.0, 1.1, and 1.2. This results in a total of over 70 unique MoA proposer mixtures. We measure the quality and diversity as follows:

- **Diversity**: We utilize the Vendi Score (Dan Friedman & Dieng, 2023) to assess the diversity among individual models in the proposer mixture. The Vendi Score represents the effective number of unique elements within a collection of samples (Dan Friedman & Dieng, 2023), with further details provided in the Appendix B.2. Specifically, for a given prompt $x$, we obtain responses from each model, denoted as $y_1, y_2, \ldots, y_6$. The diversity of the proposers for prompt $x$, denoted as $d(x)$, is calculated using the Vendi Score on the set $[y_1, \ldots, y_6]$. We then compute the overall diversity across the dataset $S$ as:

$$d = \frac{1}{|S|} \sum_{x \in S} d(x).$$

- **Quality**: We first determine the accuracy of each model on the dataset $S$, yielding values $q_1, q_2, \ldots, q_6$. The average accuracy, $q = \frac{1}{6}(q_1 + q_2 + \ldots + q_6)$, serves as our measure of the quality of the proposers. We will explore additional quality measurement strategies in later sections.

**Results**. We plot MoA's performance with corresponding diversity and quality for each mixture of proposers in Figure 2. We summarize key observations as follows:

- The trends among MMLU, CRUX, and MATH are consistently aligned.
- When the quality is fixed, increasing diversity can enhance MoA's performance.
- When the diversity is fixed, improving quality can also boost MoA's performance.
- There exists a trade-off in the achievable Pareto front between diversity and quality.
- Notably, the best performance of MoA is typically observed in the bottom right of each subplot, indicating a strong sensitivity to quality.

Previous work on ensembles (Wang et al., 2024a; Jiang et al., 2023a; Zhang et al., 2024b) primarily focuses on increasing the diversity of models within the proposer mixture. However, as shown in Figure 2, compared to Self-MoA on the best-performing model, simply aiming for greater diversity in the proposer mixture can result in lower overall quality, which may negatively impact MoA's performance. This trade-off between diversity and quality helps to explain why Self-MoA achieves superior performance across various benchmarks.

### 3.1 Statistical Analysis

To further understand the numerical correlation between MoA's performance and diversity as well as quality, we conduct linear regression for MoA's performance $t$ on diversity $d$ and quality $q$. Specifically, we fit the following equation for each dataset:

$$t = \alpha \times q + \beta \times d + \gamma, \tag{1}$$

where $\alpha, \beta, \gamma \in \mathbb{R}$ are real-valued coefficients to be determined. For each dataset, we collect around 70 data points from Figure 2 to construct the set $\{q^i, d^i, t^i\}_{i=1}^N$. The coefficients $\alpha$, $\beta$, and $\gamma$ are then derived by solving a linear regression on $\{q^i, d^i, t^i\}_{i=1}^N$. To make coefficients $\alpha$ and $\beta$ comparable, we normalize $q$ and $d$ by subtracting their means and dividing by their standard deviations (detailed in Appendix B.3), respectively. The results are presented in Table 4. We observe that the p-values for both $\alpha$ and $\beta$ are less than 0.001, indicating a significant correlation between MoA's performance and both quality and diversity (Arnold, 1990). The R-squared values from the linear regression across three datasets are approximately around 0.7, indicating that the linear model based on quality and diversity explains 70% MoA's performance and hence a strong correlation between inputs and outputs, according to Appendix B.4. In later parts, we show that using a more fine-grained quality calculation can further increase the R-square value.

**Comparing the effect strength of quality and diversity**. From Table 4, we observe that $\alpha$ is greater than $\beta$ across all three datasets. In particular, for CRUX and MATH, the gap between these two measures is even more pronounced. These results suggest that MoA's performance is particularly sensitive to variations in quality, highlighting the importance of prioritizing quality within the proposer mixture. This finding is also consistent with our observation that MoA achieves its best

Table 4: Linear regression (Equation 1) of MoA's performance $t$ on diversity $d$ and quality $q$.

| Dataset | $\alpha$ | | $\beta$ | | R-square |
|---|---|---|---|---|---|
| | Coefficient | P-value | Coefficient | P-value | |
| MMLU | $2.558 \pm 0.176$ | $< 0.001$ | $1.841 \pm 0.176$ | $< 0.001$ | 0.771 |
| CRUX | $4.548 \pm 0.459$ | $< 0.001$ | $1.421 \pm 0.459$ | $< 0.001$ | 0.685 |
| MATH | $4.719 \pm 0.416$ | $< 0.001$ | $2.839 \pm 0.416$ | $< 0.001$ | 0.760 |

Table 5: The R-square of the linear regression when we use different quality measurement methods. We find using Centered-1/K-Norm with K=2 can achieve good performance among all these three datasets.

| Dataset | Method | Average (K=1) | K=2 | K=3 | K=4 |
|---|---|---|---|---|---|
| MMLU | K-Norm | 0.771 | 0.809 | 0.832 | 0.845 |
| | Centered-1/K-Norm | 0.771 | 0.881 | 0.902 | 0.903 |
| CRUX | K-Norm | 0.685 | 0.736 | 0.765 | 0.779 |
| | Centered-1/K-Norm | 0.685 | 0.753 | 0.758 | 0.753 |
| MATH | K-Norm | 0.760 | 0.720 | 0.692 | 0.672 |
| | Centered-1/K-Norm | 0.760 | 0.720 | 0.692 | 0.672 |

performance in the bottom right of the plot in Figure 2, further supporting the effectiveness of our proposed Self-MoA approach.

**Alternative quality measurements**. We use the averaged accuracy of each individual model to measure quality in the previous analysis. In this section, we explore alternative methods for assessing the quality of proposers. Recall that $q_1, \ldots, q_6$ denote the accuracy of each individual model among proposers, and without loss of generality, we assume $q_1 \geq q_2 \geq \ldots \geq q_6$. It is reasonable to assume that the aggregator can select the correct answer from the proposers, particularly when the responses of individual models are inconsistent. In such cases, the aggregator would rely more heavily on models with better individual performance, meaning the weight of $q_1$ would be greater than that of $q_6$.

Therefore, we compare the following methods to calculate quality:

- **Average**: $\frac{1}{6} \sum_{i=1}^{6} q_i$.

- **K-Norm**: $\left( \frac{1}{6} \sum_{i=1}^{6} q_i^K \right)^{1/K}$, where a larger $K$ places more emphasis on stronger individual models.

- **Centered-1/K-Norm**: $q_1 - \left( \frac{1}{6} \sum_{i=1}^{6} (q_1 - q_i)^{1/K} \right)^K$. In this formulation, we first compute the difference between $q_i$ and the best model's $q_1$. The $1/K$ norm emphasizes the weights of models whose performance is closer to $q_1$.

All three methods are the same when $K = 1$. For each quality measurement, we fit a linear regression to assess the relationship between MoA's performance and the quality and diversity metrics, reporting the R-squared values in Table 5. Our analysis shows that in MMLU and CRUX, applying a larger weight to better-performing individual models tends to increase the R-squared values. However, this trend is inconsistent for MATH. We conjecture that this inconsistency arises because the aggregator Qwen2-7B-Instruct is relatively weak on MATH compared to the strongest individual model, Qwen2-Math-7B-Instruct. This limitation constrains the performance of MoA, leading to an inconsistent trend in the linear regression results. In contrast, on MMLU, where Qwen2-7B-Instruct is the strongest individual model, we find that the R-squared value can exceed 0.9 with $K = 2$ using the Centered-1/K-Norm. This indicates a very strong linear relationship between MoA performance and the quality and diversity metrics. Overall, we conclude that employing Centered-1/K-Norm with $K = 2$ (marked in blue) achieves strong performance across all three datasets.

Table 6: MoA of Llama-3.1-8B-Instruct and Qwen2-7B-Instruct. `l` is short for Llama-3.1-8B-Instruct and `i` is short for Qwen2-7B-Instruct.

|  | Aggregator | Proposer | MMLU |
|---|---|---|---|
| Individual | - | `i` | 66.16 |
|  | - | `l` | 66.40 |
| Mixed-MoA | `i` | `iiillll` | 70.73 |
| Self-MoA | `i` | `iiiiii` | 69.01 |
|  | `i` | `llllll` | 71.27 |

## 3.2 WHEN MIXED-MOA OUTPERFORMS SELF-MOA?

According to the quality-diversity trade-off illustrated in Figure 2, we conjecture that increasing diversity can enhance MoA's performance when the quality is controlled.

Typically, Mixed-MoA exhibits greater diversity than Self-MoA. Therefore, conditioned on similar quality, Mixed-MoA can outperform Self-MoA. This scenario arises when individual models demonstrate similar performance while still exhibiting significant cross-model diversity. For instance, if we combine three tasks of MMLU, CRUX, and MATH, the average performances of the individual models are 52.07%, 54.74%, and 50.60%, respectively (Table 3). In this combined task, each model specializes in different parts, with `i` performing best on MMLU, `d` on CRUX, and `m` on MATH.

From the "Average" column of Table 3, we observe that Mixed-MoA indeed outperforms Self-MoA of `dddddd`, which is aggregating samples from the individual model with the best average performance. Specifically, Mixed-MoA of `mmdddd` and `mmmddd` achieves the average performance of 59.86% and 60.04%, improves upon Self-MoA of `dddddd` by 0.35%. Given the reported small margin, we argue that Self-MoA is still a very competitive baseline under this setting, not to mention the dominant performance of Self-MoA over Mixed-MoA when focusing on one single task.

We further consider another single-task case on MMLU, involving two individual models: Llama-3.1-8B-Instruct and Qwen2-7B-Instruct, with Qwen2-7B-Instruct serving as the aggregator. We choose Llama-3.1-8B-Instruct because it performs similarly to Qwen2-7B-Instruct as an individual model. Table 6 demonstrates that even when the performance of two individual models is close, Self-MoA—utilizing six Llama-3.1-8B-Instruct proposers (denoted as `llllll`)—still outperforms the Mixed-MoA configuration (denoted as `iiillll`).

## 4 SCALING INFERENCE COMPUTE WITH SELF-MOA

In previous sections, we have provided evidence that Self-MoA over one strong model is straightforward but effective. As the community is becoming more aware of scaling inference time computing (Brown et al., 2024; Snell et al., 2024; Wu et al., 2024), one natural question to ask is:

*Given a strong model, does Self-MoA's performance scale with the number of repeated samples?*

Intuitively, Self-MoA cannot scale indefinitely by simply increasing the computation budget for at least three reasons:

- As more responses are sampled from a single model, the diversity among those samples tends to plateau.
- Aggregating information from many samples is more challenging for LLMs compared to handling a smaller number of samples.
- Every LLM has a context length limit (e.g., 8192 tokens for Gemma 2), which restricts the number of responses an aggregator can process at once.

While the first limitation is inherent to repeated sampling, we address the latter two by introducing Self-MoA-Seq, a sequential variant designed to manage large numbers of responses without

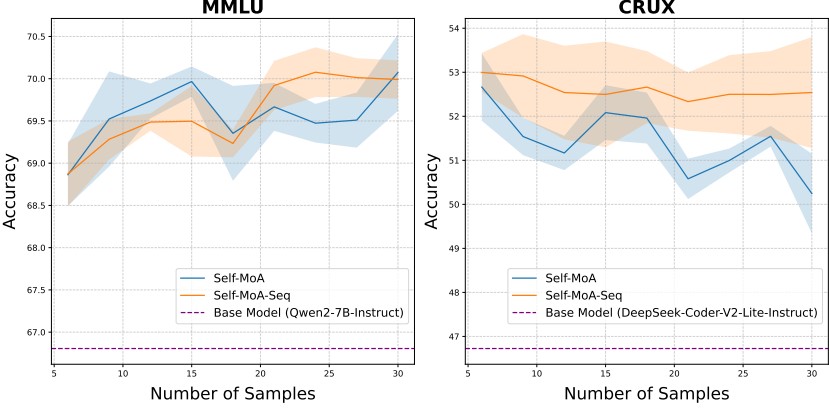

Figure 3: The performance of Self-MoA and Self-MoA-Seq with a growing number of samples. Dashed lines indicate the performance of a single forward pass with the base model.

overwhelming the aggregator. Self-MoA-Seq uses a sliding window to aggregate a fixed number of responses at a time, allowing it to handle an unlimited number of responses, regardless of context length constraints. A visual illustration is provided in Figure 1.

We evaluate the performance of Self-MoA and Self-MoA-Seq with increasing sample sizes on the MMLU and CRUX benchmarks to study their scaling behavior. For each benchmark, we use the best-performing model as both the proposer and aggregator (Qwen2-7B-Instruct for MMLU and DeepSeek-Coder-V2-Lite-Instruct for CRUX), with a sampling temperature of 0.7. In Self-MoA-Seq, the window size is set to six, with the first three slots reserved for the current synthesized output. We vary the number of samples from 6 to 30 and plot the accuracy curves from three runs with different seeds in Figure 3. Our key observations are as follows:

- Both Self-MoA and Self-MoA-Seq significantly improve performance over the individual base model.

- Adding more samples can have both positive and negative effects, meaning there is no universal compute-optimal solution.

- Self-MoA-Seq delivers performance that is comparable to, or slightly better than, Self-MoA.

These findings suggest that Self-MoA-Seq can extend the effectiveness of Self-MoA to LLMs with shorter context lengths, without sacrificing performance. Following Section 3.2, we explore whether introducing a second model can enhance performance in the sequential setting. Given that Llama-3.1-8B-Instruct performs similarly to Qwen2-7B-Instruct on the MMLU task, we compare the impact of adding Llama-3.1-8B-Instruct and DeepSeek-Coder-V2-Lite-Instruct (which underperforms Qwen2-7B-Instruct by 5%) after aggregating 30 samples from Qwen2-7B-Instruct in Self-MoA-Seq. We find that incorporating Llama-3.1-8B-Instruct boosts accuracy by around 2%, whereas adding DeepSeek-Coder-V2-Lite-Instruct reduces accuracy by more than 1.5%. This result provides another example of cross-model diversity benefiting MoA, and shows the potential of Self-MoA-Seq with increasing computation budget.

## 5 CONCLUSION

In this paper, we introduce Self-MoA, an innovative approach that utilizes in-model diversity to enhance the performance of large language models during inference. Our experiments demonstrate that Self-MoA outperforms traditional Mixed-MoA strategies in many popular benchmarks, particularly when the proposer model quality varies. By aggregating outputs from a single high-performing model, Self-MoA effectively addresses the quality-diversity trade-off. We further identify the scenarios where mixing LLM can be potentially beneficial and extend Self-MoA to the constrained context length setting. These findings highlight the potential of in-model diversity in optimizing LLM performance and pave the way for further advancements in ensemble methods.

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

## A    RELATED WORK

**Ensembles of LLMs.**    Model ensembling aims to combine strengths from multiple models. Previous studies have explored various methods to leverage a diverse set of models, including but not limited to prompting (Wang et al., 2024a), weight averaging (Lin et al., 2024; Ramé et al., 2024), routing (Jiang et al., 2024b; Lu et al., 2023), training a generative fusion model (Jiang et al., 2023b), and so on. Zhang et al. (2024a) argues that the fusion of specialized models with certain general abilities could be a promising direction toward Artificial General Intelligence. Mixture-of-Agents (MoA, Wang et al. (2024a)) first queries multiple LLMs to generate responses, then iteratively aggregates these samples through several rounds of synthesis. MoA shows promising results on several benchmarks, and its variants achieve superior performance on the AlpacaEval 2.0 leaderboard. Our method is inspired by the prompt pipeline proposed in MoA. However, while existing MoA focuses on unleashing the strength from multiple different models (Wang et al., 2024a; Jiang et al., 2023b; Zhang et al., 2024b), we demonstrate the trade-off between diversity and quality within the proposers, highlighting that focusing solely on diversity may compromise overall quality and final performance.

**LLM Inference with Repeated Sampling.**    Previous studies have shown that combining model outputs from repeated sampling can yield a better response in various domains. In tasks with automatic verifiers available, such as math (Hendrycks et al., 2021) and code (Chen et al., 2021), simply sampling LLMs multiple times can significantly improve the pass@k metric and hence boost the success rate of solving the tasks (Roziere et al., 2023; Li et al., 2022; Brown et al., 2024). In more general tasks without verification tools, we can conduct techniques like majority vote, self-consistency, and best-of-n to choose the most promising one from candidate responses (Wang et al., 2022; Chen et al., 2023b; Gui et al., 2024; Li et al., 2024). Therefore, repeated sampling is recently regarded as one approach of scaling compute during inference time (Brown et al., 2024). In this work, we identify the surprising effectiveness of repeated sampling in the context of MoA. Unlike majority vote or best-of-N, Self-MoA asks LLMs to synthesize outputs generated from repeated sampling, hence can further improve over each individual output.

**Collaborative Agents**    There is a surge of interest in building agent systems based on verification, critique, discussion, and refinement. For example, Stechly et al. (2023), Valmeekam et al. (2023), and Madaan et al. (2024) use self-critique to iteratively refine outputs through a chain structure. Madaan et al. (2024), Chen et al. (2024), and Wang et al. (2024a) explore the incorporation of multiple models to create a stronger agent that outperform each individual model. Du et al. (2023) incorporates multiple LLMs that propose and debate their individual responses over several rounds to reach a common final answer. Liang et al. (2023) proposes Multi-Agent Debate, which encourages divergent thinking during LLM debates to arrive at more informative conclusions and avoid rushing to incorrect answers. Chen et al. (2023a) introduces RECONCILE, which adopts a confidence-weighted voting mechanism for better consensus among LLM discussions. Interestingly, Wang et al. (2024b) shows that a single model with carefully designed prompts can sometimes match the performance of agent discussions. Moreover, agent discussions mainly outperform a single LLM when the prompts are insufficient.

## B    SUPPLEMENTS

### B.1    MULTI-LAYER MOA

MoA can be extended to multiple layers. For MoA with $l$ layers and $n$ LLMs $\{A_{i,j}\}_{j=1}^{n}$ in each layer $i$, we can formulate it as follows:

$$y_i = \bigoplus_{j=1}^{n} [A_{i,j}(x_i)] + x_1, \quad x_{i+1} = y_i,$$

where each LLM $A_i^j$ generates a response for the query $x_i$, which is further concatenated with the original query by the aggregator's prompt $\bigoplus$.

## B.2 VENDI SCORE

The Vendi Score (VS) is a metric designed to evaluate diversity in machine learning. It takes as input a collection of samples along with a pairwise similarity function, and it outputs a single value that represents the effective number of unique elements within the sample set.

The score is computed using a positive semi-definite similarity matrix $K \in \mathbb{R}^{n \times n}$ as follows:

$$VS(K) = \exp\left(-\mathrm{tr}\left(\frac{K}{n}\log\left(\frac{K}{n}\right)\right)\right) = \exp\left(-\sum_{i=1}^{n}\lambda_i\log(\lambda_i)\right)$$

Here, $\lambda_i$ are the eigenvalues of the normalized matrix $\frac{K}{n}$, and $0\log 0 = 0$. Essentially, the Vendi Score is the exponential of the von Neumann entropy of $\frac{K}{n}$, which reflects the Shannon entropy of its eigenvalues, also referred to as the effective rank. This metric provides a quantitative measure of diversity based on the distribution of similarity scores among the samples.

## B.3 NORMALIZATION OF INPUTS

Given a sequence of inputs $x_1, ..., x_n$. Let $x'$ denote the normalized $x$. We have

$$x' = \frac{x_i - \bar{x}}{\mathrm{std}(x)}, \text{ where } \bar{x} = \frac{1}{n}\sum_{i=1}^{n}x_i, \text{ and } \mathrm{std}(x) = \sqrt{\frac{1}{n}\sum_{i=1}^{n}(x_i - \bar{x})^2}$$

## B.4 IMPLICATION OF R-SQURE

The implications of R-squared are presented in Table 7, illustrating the degree of influence between the independent and dependent variables. (Sarjana et al., 2020).

Table 7: The interpretation of R-square

| R-square | Level |
| --- | --- |
| $[0, 0.2)$ | Very weak |
| $[0.2, 0.4)$ | Weak |
| $[0.4, 0.6)$ | Median |
| $[0.6, 0.8)$ | Strong |
| $[0.8, 1.0]$ | Very Strong |

# C ADDITIONAL RESULTS

## C.1 MT-BENCH RESULTS

We also compare MoA and Self-MoA on the MT-Bench (Zheng et al., 2023) benchmark under the same experiment setting as Wang et al. (2024a). We copy the numbers from Wang et al. (2024a) for 3-Layer MoA settings, and report our implemented results for the other experiments to ensure that 2-Layer experiments are fair comparisons. Table 8 shows that Self-MoA outperforms its Mixed-MoA counterpart, and using GPT-4o as the aggregator can achieve the best performance even with fewer forward passes compared to 3-Layer MoA with GPT-4o.

## C.2 COMPARISON TO UNIVERSAL SELF-CONSISTENCY

We conduct further experiments to compare Self-Consistency (Wang et al., 2022) with MoA and Self-MoA on the AlpacaEval 2.0 benchmark. As this benchmark is an instruction-following task

Table 8: Comparison of Self-MoA and Mixed-MoA on MT-Bench. We use Qwen1.5-110B-Chat and GPT-4o as the aggregator.

| | Model Configuration | Avg. | 1st turn | 2nd turn | # Forward Passes |
|---|---|---|---|---|---|
| Individual | WizardLM-2-8x22B | 8.99 | 9.05 | 8.93 | 1 |
| | Qwen1.5-110B-Chat | 8.61 | 8.77 | 8.45 | 1 |
| | LLaMA-3-70B-Instruct | 8.84 | 9.14 | 8.54 | 1 |
| | Qwen1.5-72B-Chat | 8.62 | 8.66 | 8.58 | 1 |
| | Mixtral-8x22B-Instruct-v0.1 | 8.49 | 8.89 | 8.09 | 1 |
| | dbrx-instruct | 7.82 | 8.21 | 7.43 | 1 |
| Mixed-MoA | 2-Layer MoA | 9.06 | 9.23 | 8.89 | 7 |
| | 2-Layer MoA w/ GPT-4o | 9.39 | 9.40 | 9.37 | 7 |
| | 3-Layer MoA | 9.25 | 9.44 | 9.07 | 13 |
| | 3-Layer MoA w/ GPT-4o | 9.40 | 9.49 | 9.31 | 13 |
| Self-MoA + WizardLM-2-8x22B | 2-Layer Self-MoA | 9.13 | 9.36 | 8.89 | 7 |
| | 2-Layer Self-MoA w/ GPT-4o | **9.52** | 9.56 | 9.47 | 7 |

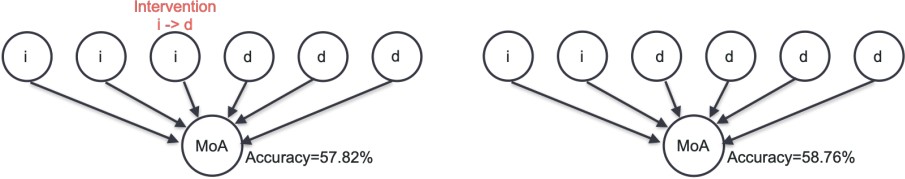

Figure 4: An illustration from a causal perspective

without exact answers, we evaluate on Universal Self-Consistency (USC) (Chen et al., 2023b) which prompts LLMs to generate the most consistent response. We report the result in Table 10, which shows that USC performs worse than its MoA counterpart when proposers and aggregators are controlled. This further suggests that rather than finding the most consistent response, MoA and Self-MoA can encourage LLM to synthesize the references and produce a better response.

### C.3    NORMALIZING SUB-TASKS IN TABLE 3

The results in Table 3 indicate that the variance of models on CRUX is generally higher than that of the other two tasks, which could bias the average performance towards CRUX. To ensure that each task contributes equally to the overall performance metric, we assign weights to the three tasks based on the inverse of their variance.

For example, considering MMLU, we report 19 performance metrics (including individual models, Mixed-MoA, and Self-MoA) in Table 3. The standard deviation of performance for MMLU across these 19 settings is calculated to be 3.50. In comparison, the standard deviation for CRUX and MATH are 5.70 and 4.27, respectively. Consequently, the weight assigned to MMLU when calculating the "WeightedAvg" is given by:

$$\text{Weight}_{\text{MMLU}} = \frac{1/3.50}{(1/3.50) + (1/5.70) + (1/4.27)}$$

The performance of weighted average is shown in Table 9.

### C.4    A DISCUSSION FROM A CAUSAL PERSPECTIVE

Consider the setting described in Table 3, where we focus on the average accuracy across three tasks. The performance of MoA is influenced by six proposers. For instance, in the combination `iiiddd`, the MoA achieves an accuracy of 57.82%. The causal graph illustrating this relationship is shown in Figure 4 Left. Now, let's examine a do intervention where we replace one instance of `i` with `d`. This changes the combination from `iiiddd` to `iidddd`, resulting in a less diverse set of

| | Aggregator | Proposer | MMLU | CRUX | MATH | Average | WeightedAvg |
|---|---|---|---|---|---|---|---|
| Individual | - | `i` | 66.16 | 36.25 | 53.81 | 52.07 | 54.46 |
| Individual | - | `d` | 60.91 | 49.51 | 53.82 | 54.74 | 55.65 |
| Individual | - | `m` | 54.36 | 27.88 | 69.57 | 50.60 | 52.80 |
| Mixed-MoA | `i` | `iimmdd` | 67.89 | 42.88 | 64.38 | 58.38 | 60.40 |
| Mixed-MoA | `i` | `imdddd` | 67.42 | 44.50 | 63.90 | 58.61 | 60.46 |
| Mixed-MoA | `i` | `iiiimd` | 68.90 | 41.25 | 63.00 | 57.72 | 59.94 |
| Mixed-MoA | `i` | `immmmd` | 66.63 | 42.75 | 66.02 | 58.47 | 60.40 |
| Mixed-MoA | `i` | `iimmmm` | 66.23 | 39.25 | 66.10 | 57.19 | 59.38 |
| Mixed-MoA | `i` | `iiimmm` | 67.49 | 38.25 | 64.16 | 56.63 | 59.00 |
| Mixed-MoA | `i` | `iiiimm` | 68.00 | 37.00 | 62.92 | 55.97 | 58.47 |
| Mixed-MoA | `i` | `iidddd` | 68.21 | 45.50 | 62.56 | 58.76 | 60.58 |
| Mixed-MoA | `i` | `iiiddd` | 68.21 | 42.88 | 62.38 | 57.82 | 59.86 |
| Mixed-MoA | `i` | `iiiidd` | 68.47 | 40.75 | 61.24 | 56.82 | 59.05 |
| Mixed-MoA | `i` | `mmdddd` | 66.34 | 46.75 | 66.48 | 59.86 | 61.45 |
| Mixed-MoA | `i` | `mmmddd` | 65.80 | 47.00 | 67.32 | 60.04 | 61.57 |
| Mixed-MoA | `i` | `mmmmdd` | 65.44 | 42.50 | 67.62 | 58.52 | 60.39 |
| Self-MoA | `i` | `dddddd` | 65.23 | 50.75 | 63.08 | 59.69 | 60.86 |
| Self-MoA | `i` | 6×TaskBest | 69.01 | 50.75 | 68.42 | 62.73 | 64.21 |
| Self-MoA | TaskBest | TaskBest | 69.01 | 52.62 | 69.80 | 63.81 | 65.14 |

Table 9: This table compares Self-MoA and Mixed-MoA using a weighted composition of three subtasks. The weights are assigned to each sub-task to prevent a high-variance task, such as CRUX, from disproportionately influencing the overall performance metrics. This approach ensures a more balanced evaluation, allowing for a fairer comparison between the two models.

Table 10: Comparison of Self-MoA, Mixed-MoA, and Universal Self-Consistency (USC) on AlpacaEval 2.0 leaderboard. We use Qwen1.5-110B-Chat as the aggregator.

| | Model Configuration | LC Win Rate | # Forward Passes |
|---|---|---|---|
| Mixed-MoA | MoA | 59.1 | 7 |
| Self-MoA | Self-MoA + WizardLM-2-8x22B | **65.7** | 7 |
| Universal Self-Consistency | Mixed-USC | 53.8 | 7 |
| | Self-USC + WizardLM-2-8x22B | 60.2 | 7 |

proposers, as it is now biased towards `d`. However, the quality of the proposers improves, since `d` is a stronger proposer in terms of average performance. This intervention demonstrates that the MoA performance increases (see Figure 4 Right), highlighting the significance of proposer quality.

