# OpenReview forum: "Rethinking Mixture-of-Agents: Is Mixing Different Large Language Models Beneficial?"
_ICLR.cc/2025/Conference — Submitted to ICLR 2025_

### Official Review · Reviewer_sa8q · 2024-10-30

**Soundness:** 2
**Presentation:** 2
**Contribution:** 2
**Rating:** 3
**Confidence:** 4

**Summary:**

The work explores the effectiveness of ensembling outputs from different LLMs or the single top-peforming model. It introduces the concepts of cross-model diversity and in-model diversity, and argues that the cross-model diversity is not essentially beneficial to the model cooperation. To this end, authors presents their method named Self-MoA which aggerates the outputs from the single top-performing model rather than the diverse models. Experiments shows the superiority of Self-MoA. Further discussions are about quality-diversity trade-off, statistical  analysis, the design of Self-MoA-Seq.

**Strengths:**

1. An seemingly reasonable method, i.e. Self-MoA and good evaluation performance.

2. Comprehensive experiments.

**Weaknesses:**

1. I think the overall technical contribution of this work is limited. The idea is likely to be a combination of self-consistency and MoA. It more likes a technical report rather than a research paper.

2. It is recommended to add the introduction of Self-MoA. Readers not familiar MoA would be confused about the oveall framework.

3. In the Section 3.1, authors model the correlations between quality and diversity with linear regression. I wonder how the authors ensure the relationship should be linear, rather than polynomial or else.

**Questions:**

See weakness.

---

> ### Author Response · Authors · 2024-11-23
> **Official Comment by Authors**
>
> Thanks a lot for reading our paper and for your insightful comments.
>
> **Q1:** Technical contribution.
> **A1:** Please refer to our general response **G1** for an overview of the technical contributions. Additionally, we would like to clarify that our proposed method differs from self-consistency. To demonstrate this distinction, we provide additional results in Appendix C.2, comparing the performance of Self-MoA, Mixed-MoA, and Universal Self-Consistency (USC) \[1\] on the AlpacaEval 2.0 benchmark.
>
> While the key difference between Mixed-MoA and Mixed-USC (or Self-MoA and Self-USC) lies in the type of prompting—MoA methods use MoA prompts, while USC methods use USC prompts—our results show that MoA approaches consistently outperform their USC counterparts. This suggests that instead of simply identifying the most consistent response, MoA and Self-MoA actively encourage the model to synthesize references, resulting in superior responses.
>
> **Q2:** Introduction of Self-MoA.
> **A2:** We have included a brief introduction to MoA in the Related Work section (Appendix A). We agree with the reviewer that providing a more detailed introduction to both MoA and Self-MoA would enhance the paper's readability. To address this, we will include a dedicated background section in the camera-ready version.
>
> **Q3:** Clarification of linear regression.
> **A3:** The primary objective of Section 3.1 is to compare the strength of the effects of quality and diversity. Linear regression is chosen as it provides a direct and intuitive statistical tool to quantify these effects. To account for potential nonlinearity, we also experimented with different quality measurements, including average, k-norm, and centered-1/k-norm, to ensure robustness in our analysis.
>
> \[1\] Chen, Xinyun, et al. "Universal self-consistency for large language model generation." *arXiv preprint arXiv:2311.17311* (2023).

---

### Official Review · Reviewer_r8rC · 2024-11-02

**Soundness:** 3
**Presentation:** 3
**Contribution:** 3
**Rating:** 6
**Confidence:** 3

**Summary:**

This paper critically evaluates the importance of diversity in an ensemble of LLM responses. It chooses the general Mixture-of-Agents (MoA) method as its target of inquiry and comprehensively evaluates it against a proposed non-diverse variant called Self-MoA which achieves superior performance. They also conduct analysis on the trade-off between model diversity in an ensemble and the quality of the outputs and find that performance is most sensitive to quality. The authors also try to identify in which cases a diversity of models could be helpful. There is also an exploration of the scalability of this method of inferencing that contributes to the discussion of scaling inference compute.

**Strengths:**

It showcases that when you know the kind of task the model will be doing, it is better to duplicate that model rather than seek diversity, which is quite an interesting finding that might go against how some currently view LLM ensembling methods.

Their argument is sound and the structure of the paper makes sense and is quite thorough in its analysis. They use creative methods to showcase the quality-diversity trade-off and supply figures that complement their findings well. I think the paper is a good contribution.

**Weaknesses:**

My main issue with this paper is that it doesn't really answer the question posed in its title very thoroughly. I think a broad question like "Is mixing different LLMs beneficial?" can be broken down to answering two questions: "When does it matter?" and "When doesn't it matter?". I feel like the paper does a very good job answering the latter question with many experiments and analyses; the former, though, isn't as well covered.

Section 3.2 attempts to tackle this question by arguing that "increasing diversity can enhance MoA's performance when the quality is controlled." I feel like this claim isn't very well supported, as there is only one case that results in an increase in average performance, and it's quite minimal, as mentioned later in the section. Moreover, using the aggregate combined task here as a metric to judge whether one configuration is better than another could be misleading as it might obscure performance variations across individual tasks. This can mask specific cases where mixing models shows benefit for some tasks but not others. I don't find it to be enough to say one model is more performant than another due to it having a higher aggregate score. I think there should be some more discussion on this.

**Questions:**

- In Table 1, what does the Individual column mean? Is it just passing the query to the model without any form of MoA? If so, what models does Mixed-MoA and 3-layer-MoA use?
The number of forward passes of 12 doesn't seem consistent to me with the fact MoA uses 5 different models per layer (so if I'm understanding this correctly it should be 3 layers * 5 models + 3 aggregations = 18 forward passes)

- In Figure 2, I assume it's a typo and the y axis of the MATH dataset is supposed to be 1.6-2.8?

Grammar issues:

- 422-423 "we fit a linear regression to..."
- 371 "explains 70% MoA’s performance"
- 444 "WHEN MIXED-MOA OUTPERFORMS SELF-MOA?"

---

> ### Author Response · Authors · 2024-11-23
> **Official Comment by Authors**
>
> Thanks a lot for reading our paper and for your insightful comments.
>
> **Q1:** Clarification of Section 3.2.
> **A1:** We agree with the reviewer that addressing both "When does it matter?" and "When doesn't it matter?" is crucial. To explore these questions, we conducted a comprehensive study across various tasks, including AlpacaEval 2.0, MT-Bench (newly added), MMLU, CRUX, and MATH. Additionally, we evaluated a large proportion of possible proposer combinations for the last three tasks. The results from these single-task evaluations consistently demonstrate that Self-MoA outperforms Mixed-MoA when one model significantly outperforms others.
>
> Building on this observation, we attempted to construct scenarios where Mixed-MoA could have an advantage:
>
> 1. Constructing a mixed task where no single model performs well across all subtasks.
> 2. Identifying two models with very similar performance on a specific task to see if combining them improves performance.
>
> We implemented both scenarios in Section 3.2 (the second is hard to construct so we only provide one case study in the last paragraph). Even in such constructed settings, Self-MoA remains a strong baseline, while Mixed-MoA provides only marginal improvements. Furthermore, in Section 4, we observe that incorporating a second model can enhance Self-MoA-Seq performance when the first model reaches saturation.
>
> Additionally, as noted in our discussion with Reviewer ajnS (see Tables 1 and 2 in our individual response to Reviewer ajnS), Mixed-MoA may be a more practical choice in certain scenarios, such as online settings where identifying the best model is difficult. For instance, Mixed-MoA shows better performance in specific categories of MT-Bench.
>
> To enhance the clarity of Section 3.2, we will incorporate these discussions into the camera-ready version.
>
> **Q2:** Clarification of Table 1\.
> **A2:** You are correct in interpreting the "Individual" column—it represents the evaluation of each model independently, without employing any form of MoA prompting. Both the 2-Layer (MoA-Lite) and 3-Layer Mixed-MoA configurations use the outputs of six individual models as proposers, with Qwen1.5-110B-Chat serving as the aggregator.
>
> For the 3-Layer MoA, the forward passes are distributed as follows: six forward passes in the 1st layer (one for each proposer), six in the 2nd layer (each proposer generates again), and one in the 3rd layer (aggregator generates once), resulting in a total of 13 forward passes.
>
> To enhance clarity, we have updated the naming conventions in Table 1 in the revised version of the paper.
>
> **Q3:** Typos in Figure 2\.
> **A3:** Thanks for spotting the typo. We have fixed it in the updated version of the paper.

---

### Official Review · Reviewer_cUzm · 2024-11-03

**Soundness:** 3
**Presentation:** 3
**Contribution:** 2
**Rating:** 3
**Confidence:** 4

**Summary:**

This paper introduces Self-MoA and surprisingly it found that it's better than MoA approach which combines outputs from multiple LLMs to enhance performance.
The authors run various experiments to use various combination of proposers and aggregators and found that using only the top-performing models to generate multiple outputs and then aggregate achieve superior performance.
Through various experiments across benchmarks, including aplacaeval 2.0, mmlu, crux and math, the authors found that focusing on quality often yields better performance than mixing diverse models, which include lower-quality outputs.
The paper also introduce Self-MoA-Seq, a sequential variant that address the limitations of aggregating outputs when the context length is a constraint.

The contributions are:
- This papers introduces Self-MoA, which outperforms Mixed MoA on various benchmarks.
- Through experiments, the authors proposed a conjecture and suggests that a core trade-off between diversity and quality can cause performance difference.
- The authors also extends Self-MoA to Self-MoA-Seq, which mitigates the context length contraint.

**Strengths:**

- The paper is well written and presented.
- Self MoA presents a practical advancement over Mixed MoA on various benchmarks.

**Weaknesses:**

- The technical novelty is limited. Compared to Mixed MoA, a main difference is exploring the tradeoff between quality and diversity. Mixed MoA already uses a few sequential layers, which is somehow related to self-moa-seq.
- There is no experiment about using larger models, e.g. GPT-4o as proposer and / or aggregator. Mixed-MoA used GPT 4o as aggregator.
- The current analysis are more of correlation analysis. Further systematic casual analysis or study on the root cause of why and when selfMoA outperforms Mixed MoA? Some studies on the aggregator's aggregation process (combining inputs from various proposers) would be helpful. e.g. attention at aggregation step.
- typo: line 155, table 2.1 -> table 1

**Questions:**

- Does selfMoA only use 1 layer (if we use the notion mentioned in Mixed MoA)
- In table 1, it seems that you use Qwen1.5-110B-Chat as the aggregator?
- Do you have results using WizardLM-2-8x22B as aggregator, as following the findings in the selfMoA paper, would this yield better results?

---

> ### Author Response · Authors · 2024-11-23
> **Official Comment by Authors (1/2)**
>
> Thanks a lot for reading our paper and for your insightful comments.
>
> **Q1:** Technical novelty.
> **A1:** Please refer to our general response **G1**.
>
> **Q2:** The connection between Mixed-MoA and Self-MoA-Seq.
> **A2:** As noted by the reviewer, the original MoA paper introduces a multi-layer structure to scale up computational capacity. However, Mixed-MoA, when applied without a sequential structure, remains constrained by context length limitations.
>
> Additionally, our Self-MoA-Seq experiments incorporate elements inspired by Mixed-MoA. Specifically, we observed that introducing a second model of similar quality (e.g., Llama-3.1-8B-Instruct) can enhance performance after the first model reaches saturation. Conversely, adding a second model of lower quality (e.g., DeepSeek-Coder-V2-Lite-Instruct) degrades performance.
>
> **Q3:** Experiments with GPT-4o.
> **A3:** We appreciate the reviewer’s suggestion to include GPT-4o in our experiments, as it further strengthens our findings. Accordingly, we report results using GPT-4o as the aggregator on the AlpacaEval 2.0 and MT-Bench benchmarks. The latter is newly added in Appendix C.1, as suggested by Reviewer ajnS.
>
> On AlpacaEval 2.0, GPT-4o as the aggregator provides a 4% performance gain for Mixed-MoA. However, Mixed-MoA with GPT-4o still falls short compared to Self-MoA, regardless of whether GPT-4o or another model is used as the aggregator or proposer. On MT-Bench, GPT-4o proves to be a stronger aggregator than the default Qwen1.5-110B-Chat, improving the average score by 0.3–0.4 for both Mixed-MoA and Self-MoA. As a result, Self-MoA with GPT-4o as the aggregator achieves the best performance overall.
>
>
> These additional results highlight that GPT-4o is a more powerful aggregator than smaller models for both Mixed-MoA and Self-MoA. More importantly, they further demonstrate the consistent effectiveness of Self-MoA compared to Mixed-MoA.
>
> **Table 1: GPT-4o results on AlpacaEval 2.0.**
>
> |  | Model Configuration | LC Win Rate | \# Forward Passes |
> | :---- | :---- | :---: | :---: |
> |  Mixed-MoA | 2-Layer MoA | 59.1 | 7 |
> |  | 2-Layer MoA  w/ GPT-4o | 63.5 | 7 |
> |  Self-MoA | 2-Layer Self-MoA \+ WizardLM-2-8x22B | **65.7** | 7 |
> |  | 2-Layer Self-MoA w/ GPT-4o \+ WizardLM-2-8x22B | 65.2 | 7 |
>
> **Table 2: GPT-4o results on MT-Bench.**
>
> |  | Model Configuration | Avg. | 1st turn | 2nd turn | \# Forward Passes |
> | :---- | :---- | :---: | :---: | :---: | :---: |
> |  Mixed-MoA | 2-Layer MoA | 9.06 | 9.23 | 8.89 | 7 |
> |  | 2-Layer MoA  w/ GPT-4o | 9.39 | 9.40 | 9.37 | 7 |
> |  Self-MoA | 2-Layer Self-MoA \+ WizardLM-2-8x22B | 9.13 | 9.36 | 8.89 | 7 |
> |  | 2-Layer Self-MoA w/ GPT-4o \+ WizardLM-2-8x22B | **9.52** | 9.56 | 9.47 | 7 |

---

> > ### Author Response · Authors · 2024-11-23
> > **Official Comment by Authors (2/2)**
> >
> > **Q4:** Causal analysis.
> > **A4:** Thank you for raising this important question regarding the causal interpretation of our findings. We present a discussion from a causal perspective as follows.
> >
> > Consider the setting described in Table 3, where we focus on the average accuracy across three tasks. The performance of the MoA is influenced by six proposers. For instance, in the combination 'iiiddd', the MoA achieves an accuracy of 57.82%. The causal graph illustrating this relationship is shown in Figure 4 Left of the revised manuscript.
> >
> > Now, let's examine a do intervention where we replace one instance of 'i' with 'd'. This changes the combination from 'iiiddd' to 'iidddd', resulting in a less diverse set of proposers, as it is now biased towards 'd'. However, the quality of the proposers improves, since 'd' is a stronger proposer in terms of average performance.
> >
> > This intervention demonstrates that the MoA performance increases (see Figure 4 Right), highlighting the significance of proposer quality. We agree that visualizing the aggregation process—such as through attention mechanisms—would provide valuable insights, and we plan to explore this in future work.
> >
> > **Q5:** Layers of Self-MoA.
> > **A5:** Using the same terminology as \[1\] (Figure 2), both Self-MoA and MoA-Lite (renamed as 2-Layer MoA in the updated version for clarity) are structured as two-layer frameworks, ensuring a fair basis for comparison in our experiments. Specifically, in both approaches, the proposers generate responses in the first layer, while the aggregator synthesizes these responses in the second layer.
> >
> > **Q6:** Qwen1.5-110B-Chat as the aggregator in Table 1\.
> > **A6:** We use Qwen1.5-110B-Chat as the aggregator in Table 1 because it is the default setting in the original paper for the AlpacaEval 2.0 benchmark (Section 3.1, \[1\]). Additionally, Table 4 in \[1\] demonstrates that Qwen1.5-110B-Chat is the strongest aggregator among the six proposers, further supporting its use in our experiments.
> >
> > **Q7:** WizardLM-2-8x22B as the aggregator in Table 1\.
> > **A7:** We experimented with using WizardLM-2-8x22B as the aggregator but found it resulted in a lower LC win rate (58.8) compared to Qwen1.5-110B-Chat (65.7). This outcome can be attributed to the fact that WizardLM-2-8x22B generates significantly longer aggregated responses (average length \= 3036\) than Qwen1.5-110B-Chat (average length \= 1923). While WizardLM-2-8x22B achieves a higher raw win rate (79.1) compared to Qwen1.5-110B-Chat (61.5), it incurs a greater penalty from the length-control debiasing mechanism in AlpacaEval 2.0, ultimately resulting in a lower LC win rate.
> >
> > \[1\] Wang, Junlin, et al. "Mixture-of-Agents Enhances Large Language Model Capabilities." *arXiv preprint arXiv:2406.04692* (2024).

---

> > ### Comment · Reviewer_cUzm · 2024-11-24
> >
> > regarding to A1, A2.
> > Thank you for your response. That's helpful.
> >
> > regarding to A3.
> > Can you please clarify what "2-Layer Self-MoA w/ GPT-4o + WizardLM-2-8x22B" means?
> > What are the proposers and aggregators?
> > For my question, I was mainly wondering if this finding scale up to larger models. e.g. only use gpt4o as proposers and aggregators.

---

> ### Author Response · Authors · 2024-11-24
> **Clarification to A3**
>
> Thank you for your prompt and insightful feedback!
>
> For "2-Layer Self-MoA w/ GPT-4o + WizardLM-2-8x22B," we used WizardLM-2-8x22B as the proposer and GPT-4o as the aggregator. Similarly, for "2-Layer MoA w/ GPT-4o," we employed six individual models from the Mixed-MoA setting as proposers, with GPT-4o serving as the aggregator.
>
> Following your suggestion, we conducted experiments to evaluate the setting where GPT-4o is used as both the proposer and the aggregator. The updated results are presented below. Interestingly, in two tasks, "2-Layer Self-MoA w/ GPT-4o + GPT-4o" did not achieve the best performance. This suggests that GPT-4o may not serve as an optimal aggregator for its own outputs. A similar observation was also noted in **A7**, where WizardLM-2-8x22B was not the most effective aggregator for its own outputs. Selecting the most effective aggregator for a given set of proposers remains an intriguing direction for future research.
>
> **Table 1: GPT-4o results on AlpacaEval 2.0.**
>
> |  | Model Configuration | LC Win Rate | \# Forward Passes |
> | :---- | :---- | :---: | :---: |
> |  Mixed-MoA | 2-Layer MoA | 59.1 | 7 |
> |  | 2-Layer MoA  w/ GPT-4o | 63.5 | 7 |
> |  Self-MoA | 2-Layer Self-MoA \+ WizardLM-2-8x22B | **65.7** | 7 |
> |  | 2-Layer Self-MoA w/ GPT-4o \+ WizardLM-2-8x22B | 65.2 | 7 |
> |  | 2-Layer Self-MoA w/ GPT-4o \+ GPT-4o | 65.3 | 7 |
>
> **Table 2: GPT-4o results on MT-Bench.**
>
> |  | Model Configuration | Avg. | 1st turn | 2nd turn | \# Forward Passes |
> | :---- | :---- | :---: | :---: | :---: | :---: |
> |  Mixed-MoA | 2-Layer MoA | 9.06 | 9.23 | 8.89 | 7 |
> |  | 2-Layer MoA  w/ GPT-4o | 9.39 | 9.40 | 9.37 | 7 |
> |  Self-MoA | 2-Layer Self-MoA \+ WizardLM-2-8x22B | 9.13 | 9.36 | 8.89 | 7 |
> |  | 2-Layer Self-MoA w/ GPT-4o \+ WizardLM-2-8x22B | **9.52** | 9.56 | 9.47 | 7 |
> |  | 2-Layer Self-MoA w/ GPT-4o \+ GPT-4o | 9.26 | 9.22 | 9.31 | 7 |
>
> We hope our reply addresses all your questions and will be happy to have further discussions if you have additional questions or comments.

---

### Official Review · Reviewer_ajnS · 2024-11-04

**Soundness:** 1
**Presentation:** 2
**Contribution:** 2
**Rating:** 3
**Confidence:** 4

**Summary:**

This paper proposes a revision to the Mixture-of-Agents approach, which first samples responses from multiple diverse LLMs / agents (proposers), before feeding it into a final LLM (an aggregator) to generate a final response given the other LLM's responses. The paper argues that the prioritization of diversity of proposer LLMs in the response generation phase increases the utilization of low quality models, thereby hurting overall MoA quality.

The paper then proposes Self-MoA: instead of sampling k total proposals from n different LLMs, one can sample k proposals from the best overall LLM. By doing so, Self-MoA outperforms standard MoA on the same model settings and benchmarks. When applied to SoTA open source models on AlpacaEval 2.0 leaderboard, Self-MoA achieves SoTA, setting a new high score in the public leaderboard. To motivate their change, the paper performs an empirical study of the trade-off between proposal quality, proposal diversity, and end MoA quality, in a controlled setting. The study is evaluated on MMLU, CRUX (Code), and MATH, considering proposals from models that are experts in each respectively. The paper finds that while both proposal quality and diversity correlate with end quality, there is a trade-off between them, and the strongest end-MoA quality results prioritize quality. Thereby motivating their method.

Finally the paper introduces a variant of Self-MoA , called Self-MoA-Seq, that allows for unconstrained MoA scaling w.r.t. the aggregator model's context-length. It does this by breaking up the large number of proposals over multiple rounds of aggregation. They find that Self-MoA-Seq maintains or improves upon the quality of Self-MoA, without requiring as long as a context length. They however, do not find that Self-MoA responses necessarily get better with increased sampling scaling.

**Strengths:**

Originality & Significance:
- This paper is the first to thoroughly study the trade-off between the quality and diversity of proposer LLMs in the Mixture-of-Agents framework. Importantly, by doing this analysis, they find that the existing MoA approach may overemphasize diversity, regardless of proposer quality, and in fact low proposer quality can negatively impact end MoA quality. (Among other interesting findings.)
- They propose an effective revision to MoA: use only the best agent but sample multiple diverse responses (up to the same sampling budget as MoA). They demonstrate that on a major leaderboard like AlpacaEval 2.0 this variant outperforms standard MoA and sets a new record / all time SoTA result. This is a very significant result.
- Generally this finding would be important to the community, to highlight the possible pitfalls of doing MoA, and serve as a deep dive into the specific design decisions to consider when applying such an approach.

Quality:
- The paper presents a thorough study of the tradeoffs between different compositions of proposing agents in the MoA process.
- The methodology is well controlled to be directly comparable with MoA, and the baselines are strong.

Clarity: The paper is clearly written and laid out. It is easy to read and follow, free of writing or grammatical errors.

**Weaknesses:**

1. Soundness. I have major soundness concerns regarding how the authors interpret the results of their experiments. Regarding Table 3, on lines 261 to 265, the authors write that only two configurations (of proposer compositions) of MoA slightly outperform / match Self-MoA, with the rest underperforming Self-MoA. However, the authors here take the overall average at face value for this comparison.
- When looking at this "best Self-MoA" configuration, "dddddd", this configuration is the worst on MMLU, and in the bottom half for MATH. The overall average of this method only beats the others since it appears that CRUX (Coding) shows the most variance in scores from MoA configurations. Allocating all samples to 'd' improves coding the most, and the outlier coding result boosts the overall average of Self-MoA.
- The basic MoA, that allocates sampling budget equally across agents "iiddmm" beats Self-MoA "dddddd" on 2 of 3 capabilities.
- The best results in the table are the TaskBest results. However it is unclear to me how TaskBest is "Self-MoA" rather than "Mixed MoA" as it still uses a mixture of all the agents.

In the end, the paper asks the overarching question "Is mixing different large language models beneficial?" and the paper's Table 3 essentially does show: yes it is, although the authors do not acknowledge this. The value of MoA lies in the ability to potentially construct a well rounded model from multiple different experts. In Table 3 every row that uses multiple models have a better capability balance than the author's proposed single-model method "dddddd", which excels only in Code and uses this narrow ability to bolster the overall average. TaskBest results here directly disproves the single-model Self-MoA proposal of the paper, showing that the best overall results come from deciding when to use each of the expert models.

2. Soundness. A counter argument to my concern above might be Table 2, where the authors are able to boost a single general model to #1 on AlpacaEval 2.0. While this is true, AlpacaEval represents a specific evaluation setting and does not necessarily imply that there is no value to mixing multiple models. This #1 result also uses a stronger base model that was not tried by the original MoA method. This result then, while impressive, does not provide insight on whether Self-MoA is better than Mixed-MoA.
- In Table 1, a more fair comparison is done. While Self-MoA improves over 3-layer MoA by 0.3 win rate, this small amount is certainly within the margin of noise for win rate for any autorater. The original MoA paper reports the noise for the eval to be +/- 0.6. It is fair to say however that Self-MoA is able to save some compute in the process.
- I will also note that in Table 1, the use of "forward pass" is incorrect as decoding a generation requires multiple forward passes. The authors seem to mean just the number of proposals / generations required.

Overall for the authors to be able to claim that Self-MoA is better than Mixed-MoA in terms of end-quality on challenging real benchmarks, the authors would need a fair comparison, and to show a larger margin of gain given the autorater noise. This comparison would benefit from using a second benchmark also other than AlpacaEval 2.0. Such as MTBench, as shown in the original MoA paper.
It would also be important to show that the benchmark hacking raised in (1) doesn't happen for AlpacaEval 2.0 / MTBench -- that the method improves on similar prompt categories as Mixed-MoA and does not overfit to one particular category.

4. Presentation:
- Section 4 feels a bit out of place for the paper. Solving the ability to scale MoA to arbitrary number of samples seems to be a different problem altogether. It is unclear why the Self-MoA-Seq technique couldn't be applied to Mixed-MoA, and where there isn't a comparison with Mixed-MoA in the inference scaling experiment. In this setting I would expect Mixed-MoA to improve more with more samples. However, the authors do not provide this comparison.
- The organization of the paper is a bit fragmented. Starting with the proposed method, then empirical analysis of the problem, then extensions to longer context lengths.

**Questions:**

Did you ever run the Figure 3 inference scaling experiments with Mixed-MoA? It seems like as only the proposals differ, the approach should still be applicable.

---

> ### Author Response · Authors · 2024-11-23
> **Official Comment by Authors (1/2)**
>
> Thanks a lot for reading our paper and for your insightful comments.
>
> **Q1:** Soundness of Table 3\.
> **A1:** Please refer to our general response **G2**.
>
> **Q2:** Soundness of Table 1 and Table 2\.
> **A2:** We agree with the reviewer that “times of generations” could be a more precise metric than “forward pass.” However, we would like to clarify that the comparison between Self-MoA and MoA in Table 1 is indeed fair. Specifically, Self-MoA \+ WizardLM-2-8x22B and MoA-Lite (as named in \[1\]) share an identical inference pipeline. Both methods employ a two-layer framework: proposers generate responses in the first layer, while the aggregator synthesizes responses in the second layer.
>
> As both approaches utilize Qwen1.5-110B-Chat as the aggregator, the sole distinction lies in the proposers. Self-MoA uses only WizardLM-2-8x22B, whereas MoA-Lite (2-layer MoA) relies on all six models listed in the “Individual” section of Table 1\. Given that 2-layer Self-MoA significantly outperforms 2-layer MoA, it is reasonable to conclude that ensembling multiple outputs from a single model is more effective than ensembling outputs from multiple models.
>
> To improve clarity, we have updated the naming in Table 1 for better readability in the revised version.
>
> Building on the findings from Table 1, we conducted the experiments presented in Table 2 to further explore and demonstrate the potential of Self-MoA. In this set of experiments, we selected state-of-the-art models as proposers and aggregators and reported the corresponding results. As the reviewer correctly noted, the purpose of Table 2 is not to directly compare Self-MoA and Mixed-MoA, as that comparison has already been addressed in Table 1, as discussed earlier.
>
> **Q3:** Comparison on MT-Bench.
> **A3:** We agree with the reviewer that providing additional results on MT-Bench adds value to our work. Please refer to Table 8 in Appendix C.1 for a detailed comparison. As discussed in A2, a fair comparison is between 2-layer MoA and 2-layer Self-MoA, both using Qwen1.5-110B-Chat or GPT-4o as the aggregator. The results demonstrate that Self-MoA consistently outperforms its Mixed-MoA counterpart, with the best performance achieved when GPT-4o is used as the aggregator. Notably, this is accomplished with less computation compared to a 3-layer MoA configuration using GPT-4o.
>
> While the performance improvements are relatively modest, as current single models already achieve high win rates on MT-Bench (as suggested in \[1\]), these results further validate the effectiveness and efficiency of the Self-MoA approach.

---

> > ### Author Response · Authors · 2024-11-23
> > **Official Comment by Authors (2/2)**
> >
> > **Q4:** Benchmark hacking.
> > **A4:** To demonstrate that Self-MoA is not overfitted to a specific category, we present a detailed comparison of the win rates for 2-layer Self-MoA and 2-layer Mixed-MoA across fine-grained categories of AlpacaEval 2.0 and MT-Bench in the table below. The results indicate that on AlpacaEval 2.0, Self-MoA achieves higher win rates across all subsets of different categories. On MT-Bench, Self-MoA achieves a similar or higher win rate on six out of eight benchmarks, as the best overall model (WizardLM-2-8x22B) can be less performant in some specific tasks such as coding. The consistent trend provides compelling evidence that Self-MoA is not overfit to one particular category, but performs consistently better than Mixed-MoA.
> >
> > **Table 1: Fine-grained win rate of Self-MoA and Mixed-MoA on AlpacaEval 2.0. We report win rate rather than length-controlled win rate as the former is more reliable for fine-grained analysis.**
> >
> > |  | [hh-rlhf](https://huggingface.co/datasets/Anthropic/hh-rlhf/viewer/Anthropic--hh-rlhf/test) | [koala](https://github.com/arnav-gudibande/koala-test-set) | [open-assistant](https://huggingface.co/datasets/OpenAssistant/oasst1/viewer/OpenAssistant--oasst1/validation) | [self-instruct](https://github.com/yizhongw/self-instruct) | [vicuna](https://lmsys.org/blog/2023-03-30-vicuna/) | overall |
> > | :---- | :---: | :---: | :---: | :---: | :---: | :---: |
> > | Mixed-MoA | 0.57 | 0.56 | 0.51 | 0.61 | 0.55 | 0.56 |
> > | Self-MoA | 0.62 | 0.60 | 0.59 | 0.65 | 0.59 | 0.62 |
> >
> > **Table 2: Fine-grained win rate of Self-MoA and Mixed-MoA on MT-Bench.**
> >
> > |  | coding | extraction | humanities | math | reasoning | roleplay | stem | writing | overall |
> > | :---- | :---: | :---: | :---: | :---: | :---: | :---: | :---: | :---: | :---: |
> > | Mixed-MoA | 8.20 | 9.35 | 9.90 | 7.35 | 8.40 | 9.50 | 10.00 | 9.75 | 9.06 |
> > | Self-MoA | 7.40 | 9.52 | 10.00 | 8.25 | 8.80 | 9.65 | 9.90 | 9.50 | 9.13 |
> >
> > **Q5:** Combine Self-MoA-Seq with Mixed-MoA.
> > **A5:** Combining the Self-MoA-Seq technique with Mixed-MoA is an interesting idea, however, this is different from the original MoA paper which proposes a multi-layer structure to scale up the compute. We would also like to point out that our Self-MoA-Seq experiments already incorporate a concept from Mixed-MoA. Specifically, we observed that introducing a second model of similar quality (e.g., Llama-3.1-8B-Instruct) can enhance performance after the first model reaches saturation. Conversely, adding a second model of lower quality (e.g., DeepSeek-Coder-V2-Lite-Instruct) degrades performance.
> >
> > \[1\] Wang, Junlin, et al. "Mixture-of-Agents Enhances Large Language Model Capabilities." *arXiv preprint arXiv:2406.04692* (2024).

---

> ### Comment · Reviewer_ajnS · 2024-11-28
>
> I thank the authors for providing additional MT Bench results. It does add more evidence that the approach works for chat applications.
>
> However, my main concern about this paper stands: "In the end, the paper asks the overarching question "Is mixing different large language models beneficial?" and the paper's Table 3 essentially does show: yes it is, although the authors do not acknowledge this. The value of MoA lies in the ability to potentially construct a well rounded model from multiple different experts."
>
> Regarding G2, while yes the average does correspond to average performance when getting a random task from some stream of queries equally weighted, "dddddd" is still the weakest model for MMLU and on the bottom end for MATH. "dddddd" loses in 2 of 3 tasks. This is true in the new variance normalized Table 9 results also. I do not think an overall average is sufficient to claim that MoA is not needed. This is not just a theoretical thing: in real production settings, the worst loss bucket can represent the most embarrassing losses. MoA offers a tool for addressing specific loss buckets via ensembling.
>
> For example, I can come up with a simple example where Self-MoA would fail: three models, A only does coding and nothing else, B only does math and nothing else, C only answers biology questions and nothing else. In this case there is no good singular model choice for Self-MoA. Are there some real world settings where it is beneficial to select one model to sample from? Absolutely! The authors show this very well. However, this is not sufficient to claim the generality of Self-MoA as a method.
>
> In addition, the "TaskBest" results (the strongest results) clearly proves that "mixing different large language models [*is*] beneficial" as it aggregates outputs from multiple LLMs. This essentially directly contradicts the abstract, the main thesis of the paper.
>
> I think this paper has value to the community, and offers interesting insights, but as it stands the framing of the work tries to sell Self-MoA over MoA when it doesn't have to. In the process, this framing introduces claims that are unsound, and I cannot raise my score at this time. The paper instead could have served as a valuable analysis of MoA and potential improvements to MoA (e.g. *when* is it better to sample from a single model, TaskBest being some sort of Routed MoA, etc). I hope the authors consider this for future revisions (not necessarily ICLR).

---

> > ### Author Response · Authors · 2024-12-03
> >
> > We thank the reviewers for the reply. Here we provide further clarifications to address the reviewer’s concern.
> >
> > As the reviewer r8rC suggested, to answer "Is mixing different LLMs beneficial?", we aimed to answer two questions — "When does it matter?" and "When doesn't it matter?" To explore these questions, we conducted a comprehensive study across various tasks, including AlpacaEval 2.0, MT-Bench (newly added), MMLU, CRUX, and MATH. Additionally, we evaluated a large proportion of possible proposer combinations for the last three tasks. The results from these single-task evaluations consistently demonstrate that Self-MoA outperforms Mixed-MoA when one model significantly outperforms others.
> >
> > We fully agree with the reviewer’s concern that Self-MoA should not always be the best choice, which is precisely the point of Section 3.2. In that section, we deliberately constructed scenarios where Mixed-MoA could have an advantage:
> > 1. Constructing a mixed task where no single model performs well across all subtasks.
> > 2. Identifying two models with very similar performance on a specific task to see if combining them improves performance.
> >
> > We implemented both scenarios in Section 3.2 (the second is hard to construct so we only provide one case study in the last paragraph). Even in such constructed settings, Self-MoA remains a strong baseline, while Mixed-MoA provides only marginal improvements. Furthermore, in Section 4, we observe that incorporating a second model can enhance Self-MoA-Seq performance when the first model reaches saturation.
> >
> > Additionally, as noted in our previous discussion (see Tables 1 and 2), Mixed-MoA may be a more practical choice in certain scenarios, such as online settings where identifying the best model is difficult. For instance, Mixed-MoA shows better performance in specific categories of MT-Bench.
> >
> > To enhance the clarity of Section 3.2, we will incorporate these discussions into the camera-ready version.

---

### Author Response · Authors · 2024-11-23
**General Response**

We sincerely thank all the reviewers for their constructive feedback and for recognizing our work as a "comprehensive, thorough, and creative study" of the quality-diversity trade-off within the MoA framework (Reviewers ajnS, r8rC, and sa8q). We appreciate the acknowledgment of our manuscript as "well written" (Reviewer cUzm) and for highlighting the "interesting and significant results" we present (Reviewers ajnS and r8rC).

Here we respond to all major concerns raised by several reviewers, and reply to reviewers’ other questions individually under their comments. Following Reviewer ajnS's suggestion, we have included new experimental results on MT-Bench in the revised version of the paper (Appendix C.1, highlighted in red), which further reinforce our findings.

**G1: Technical Novelty and Contribution.**

We appreciate Reviewers ajnS and r8rC for acknowledging our work as “the first to thoroughly study the trade-off between the quality and diversity of proposer LLMs in the Mixture-of-Agents framework,” employing “creative methods” to uncover “interesting and important findings” and introducing “an effective revision to MoA” that achieves “a very significant result.” We would like to stress that identifying and addressing potential limitations of existing approaches is as critical, if not more so, than devising entirely new algorithms.

In this spirit, we conduct a comprehensive analysis of MoA and Self-MoA across a diverse set of tasks, including AlpacaEval 2.0, MMLU, CRUX, MATH, and the newly added MT-Bench. This analysis verifies the quality-diversity trade-off and demonstrates the effectiveness of Self-MoA. Our findings contribute new insights into the MoA framework and, more broadly, the application of model ensembling. Specifically, we highlight the importance of balancing diversity and quality in ensemble design, emphasizing that both aspects are pivotal to achieving robust performance.

**G2: Interpretation of Table 3.**

To interpret Table 3, we analyze the results using two perspectives: focusing on individual tasks (sub-task aware) and considering a mixture of all three tasks (sub-task agnostic).

(A) Sub-task Aware Perspective

This approach evaluates each task individually, allowing us to select the most suitable MoA scheme for each task. For example, model “i” (Qwen2-7B-Instruct) performs best on the MMLU task, while model “d” (DeepSeek-Coder-V2-Lite-Instruct) excels on the CRUX task. These models are referred to as the task-best proposers, with Self-MoA configurations for MMLU and CRUX being “iiiiii” and “dddddd,” respectively.

When we fix the proposer as “i”, it becomes evident that Self-MoA utilizing the task-best proposer consistently achieves higher performance compared to MoA with mixed proposers across all tasks. It is important to note that in this context, we analyze each task separately, demonstrating that Self-MoA consistently outperforms each mixture combination. Specifically, the performance improvements of Self-MoA with the task-best proposer over the best Mixed-MoA are as follows: MMLU shows an improvement of 0.11%, CRUX improves by 3.75%, and MATH increases by 0.80%.

Notably, the best Mixed-MoA configurations are selected independently for each task from 13 combinations; for example, the optimal Mixed-MoA for MMLU is “iiiimd,” while for CRUX it is “mmmddd.” This indicates that, for each task considered individually, Self-MoA, which prioritizes task quality, outperforms previous MoA approaches that focus on diversity.

(B) Task Mixture (Sub-task Agnostic Perspective)

Now, consider a scenario where we have a dataset that is an equally weighted mixture of these three tasks. In this case, our model may encounter any testing sample from any of the three tasks, and we do not know the origin of each testing sample. Consequently, we can only employ a fixed MoA strategy for testing samples drawn from the three tasks with equal probability. Given that model “d” is the strongest in this mixed context, we use “dddddd” for Self-MoA.

From the “Average” column in Table 3, we observe that even in this challenging scenario—where each individual model excels on a subset of the data—the best Mixed-MoA out of 13 candidates only marginally outperforms Self-MoA by 0.35%. This small margin suggests that Self-MoA remains a highly competitive baseline in this setting, especially considering its dominant performance over Mixed-MoA when focused on individual tasks.

**Imbalanced Variance:** The results in Table 3 indicate that the variance of models on CRUX is generally higher than that of the other two tasks, which could bias the average performance towards CRUX. To ensure that each task contributes equally to the overall performance metric, we assign weights to the three tasks based on the inverse of their variance. In Table 9 of the revised version. The trend and conclusions are similar.

Therefore, we think that the conclusion “Self-MoA works better than MoA consistently” holds consistently.

---

> ### Comment · Reviewer_cUzm · 2024-11-24
> **Can you highlight the changes in the paper**
>
> There are various changes in the paper, can you please highlight (using a different color) what are changed in the paper, compared to the version that was reviewed.
> e.g. Table 1, "Self-MoA + WizardLM-2-8x22B" seems to be changed to "2-Layer Self-MoA + WizardLM-2-8x22B". This is fine, but I am not sure if there are any other changes.

---

> ### Author Response · Authors · 2024-11-24
> **Thanks for your suggestion!**
>
> Thank you for your valuable feedback and suggestion. We updated Table 1's naming in the main paper to improve clarity as suggested by reviewers. All other changes have been highlighted in red in the Appendix. Additionally, we have now specifically highlighted the change to Table 1 in the updated version for improved transparency. We appreciate your input and hope this addresses your concern.

---

### Meta-Review · Area_Chair_crZS · 2024-12-21

**Metareview:**

This paper proposed an alternative to mixture of agents by simply sampling multiple samples from the single top language model. This paper studies the tradeoff between proposal diversity and quality, and found that existing approaches might overemphasize diversity over proposal quality, and that simply sampling from the top model works better across many chat benchmarks such as AlpacaEval and MT-Bench.

Strengths:
The proposed approach of sampling from the single-best model sets a new SOTA on AlpacaEval and challenges the current standard practice of ensembling multiple language models, which is a significant result.

Weaknesses:
As pointed out by multiple reviewers, the research boils down to two questions: when does mixing models matter? When doesn't it matter? While the latter is answered well, the former isn't. Also, one reviewer pointed out that the results in Section 3.2 show that the proposed method might not always help and therefore lacks generality.

Overall, I think this paper proposes an alternative to mixture of agents that shows a surprising and interesting result. However, reviewers are concerned about the novelty of the proposed approach and the claims made that seem not consistent with the results. I'd recommend authors to change the writing to make it clear that the claim is not that the proposed approach is better, just instead when it's better and when it's not. For now I'm recommending reject based on reviewer scores, although I wouldn't mind if this paper gets accepted.

**Additional Comments On Reviewer Discussion:**

There are two main concerns: one is that the proposed approach is not always doing better over mixture of agents based on results, but the authors have clarified that the setting was constructed to show when the proposed approach doesn't work. I think the confusion might arise from the way the abstract was written, and changing it might clarify the intention of this work. Another concern is the novelty of the proposed approach, but I don't consider that a big problem since the empirical results are surprising and interesting.

---

### Decision · Program_Chairs · 2025-01-22

Reject